# SELF-TAUGHT EVALUATORS

## ABSTRACT

Model-based evaluation is at the heart of successful model development – as a reward model for training, and as a replacement for human evaluation. To train such evaluators, the standard approach is to collect a large amount of human preference judgments over model responses, which is costly and the data becomes stale as models improve. In this work, we present an approach that aims to improve evaluators *without human annotations*, using synthetic training data only. Starting from unlabeled instructions, our iterative self-improvement scheme generates contrasting model outputs and trains an LLM-as-a-Judge to produce reasoning traces and final judgments, repeating this training at each new iteration using the improved predictions. Without any labeled preference data, our Self-Taught Evaluator can improve a strong LLM (Llama3-70B-Instruct) from 75.4 to 88.3 (88.7 with majority vote) on RewardBench. This outperforms commonly used LLM judges such as GPT-4 and matches the performance of the top-performing reward models trained with labeled examples.

## 1 INTRODUCTION

Large language models (LLMs) rely on strong evaluators at every stage of the development lifecycle. They are used at training time as reward models to align with human preferences (Bai et al., 2022; Ouyang et al., 2022) or for iterative self-improvement (Yuan et al., 2024), and at inference time as an alternative to human evaluation (Li et al., 2023; Chiang & Lee, 2023; Wang et al., 2023a; Liu et al., 2023). Improvements in evaluation capabilities will thus clearly benefit this entire workflow – including empowering the scientific research process itself as we aim to develop better overall techniques.

Building such strong evaluator models usually relies on large amounts of high-quality preference data from human annotation over model responses, which can be costly and time-consuming to collect, as it requires expert annotation for challenging tasks (e.g., coding and mathematics). This dependency on human-generated data poses significant challenges for scaling to new tasks or evaluation criteria. Furthermore, as new models inevitably improve over older ones, these existing annotations will typically become outdated, as the judgments are based on annotations of older, less performant, model responses.

In this work, we instead explore an iterative self-training approach (Figure 1) which uses *no human annotated preferences* in the training loop, relying purely on synthetically generated data. Given a seed model, our method first uses prompting to generate contrasting synthetic preference pairs for a given input, such that one response is designed to be inferior to the other. Next, using the model as an LLM-as-a-Judge, we generate reasoning traces and judgments for these pairs, which we can label as correct or not given our synthetic preference pair design. After training on this labeled data we obtain a superior LLM-as-a-Judge, from which we can then iterate the whole process in order for it to self-improve.

In our experiments, starting from Llama-3-70B-Instruct, the proposed method improves the accuracy on RewardBench (Lambert et al., 2024) from 75.4 to 88.7 (with majority vote, or 88.3 without). This matches or outperforms the performance of reward models derived from the same Llama-3-70B-Instruct model that uses human annotations, for example using the HelpSteer2 dataset (Wang et al., 2024c) of 10k annotations achieves 85.6 using the same LLM-as-a-Judge setup.

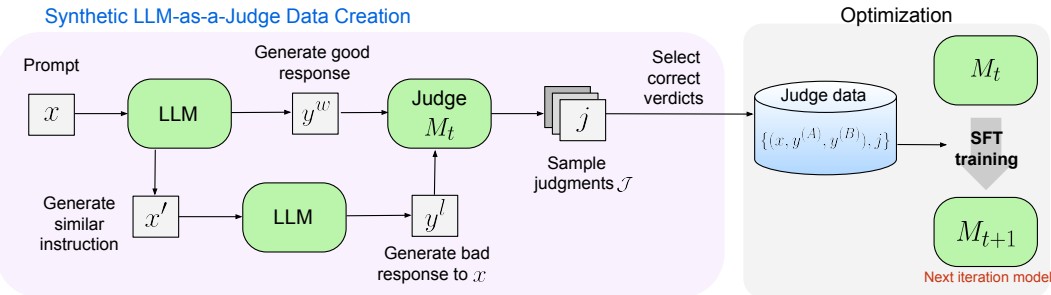

Figure 1: **Self-Taught Evaluator iterative training scheme.**

## 2 RELATED WORK

**LLM-based Evaluators**   While traditional evaluation benchmarks employ automated metrics that require a reference answer (Wang et al., 2019; Rajpurkar et al., 2016), these types of benchmarks can pose severe limitations when evaluating open-ended or complex instructions where multiple valid answers are possible (e.g., creative writing and coding). Because human evaluation per response can be costly, many recent works have proposed LLMs as effective evaluators. These come in several flavors: as classifiers that output scores directly (Zhu et al., 2023; Wang et al., 2024a) or via *LLM-as-a-Judge* prompting that can first generate a chain-of-thought in natural language, which helps provide explanations for judgments (Zheng et al., 2023). Responses can also be scored alone (Kim et al., 2023) or pairwise relative to each other (Dubois et al., 2023; Li et al., 2023; Bai et al., 2023; Saha et al., 2024). LLM evaluation shows great promise as a scalable proxy for human raters, and in the case of LLM-as-a-Judge as an explainable proxy as well (Ye et al., 2024; Zheng et al., 2023). However, many of these "off-the-shelf" evaluators demonstrate high variance across many tasks (Bavaresco et al., 2024), indicating the need for improved methods.

**Synthetic Data**   Synthetic data has emerged as a promising solution for efficiently acquiring training examples and can be particularly valuable in settings where real-world data can be hard to access (e.g., weather data covering all conditions (Lam et al., 2023)) or where correct annotations can be challenging to acquire (e.g., coding tasks (Liu et al., 2024)). Additionally, synthetic data has the benefit of being easily customizable to specific requirements, such as different evaluation criteria or safety constraints (Kim et al., 2023; El Emam et al., 2020; Howe et al., 2017). The use of synthetic data has been beneficial in model alignment (Lee et al., 2023), improving the original model's capabilities (Yuan et al., 2024; Li et al., 2024a; Yu et al., 2024; Li et al., 2024b), and teaching the model new skills (Schick et al., 2023; Lanchantin et al., 2023). In the context of evaluation, synthetic data has been used to measure tasks such as factuality (Wei et al., 2024; Feng et al., 2023), safety (Perez et al., 2023; Hubinger et al., 2024), coding (Gu et al., 2024), and general instruction following (Zeng et al., 2024), showing strong correlation with real human judgments. The West-of-n approach (Pace et al., 2024) has been used to improve reward models by constructing preference pairs using the best and worst scoring pairs from an initial model. For LLM-as-a-Judge models specifically, synthetic responses have been generated by prompting the LLM to produce a given quality response (Kim et al., 2023).

## 3 METHOD

We consider the setting of pairwise evaluation using the LLM-as-a-Judge approach (Zheng et al., 2023) that takes:

- an input (user instruction) $x$; and
- two possible assistant responses $y^{(A)}$ and $y^{(B)}$ to the user instruction $x$; and
- the evaluation prompt containing the rubric and asking to evaluate and choose the winning answer, see e.g., Figure 8.

The goal of the LLM-as-a-Judge model is to output a preference of which response $y$ is better: A or B. In order to do this it is common to output, prior to the final judgment, a chain-of-thought (or

---

**Prompt Template for Generating Response Pairs with Synthetic Preference**

Below is a conversation between an user and an AI Assistant.

{Instruction}

The start of Assistant's Answer
{Baseline Response}
The end of Assistant's Answer

Please first generate a modified instruction that is highly relevant but not semantically identical to the instruction above from the user. Then write a high-quality answer which is a good response to the modified instruction but not a good response to the original user question. IMPORTANT: Please strictly follow the following format:

User Question Modified
<provide a modified instruction here>

The start of Assistant's answer to the modified instruction
<provide a high-quality response to the modified instruction>
The end of Assistant's answer to the modified instruction

---

Figure 2: **Generating Synthetic Response Pairs.** We use the following prompt template which is used to generate a "worse response" $y^l$. Given an instruction $x$ and baseline response $y^w$ generated by an instruction-following LLM as usual, this prompt is used to first generate a "noisy" version $x'$ of the original instruction $x$, and then a best-attempt $y^l$ at responding to $x'$. $y^l$ is then treated as a poor response to $x$, giving a preference pair $y_i^w \succ y_i^l$.

"reasoning chain"), which is a set of steps generated in natural language that helps the model decide its final judgment.

Such models can be used as *pairwise reward models* to build training data for preference optimization, e.g., for training methods like DPO (Rafailov et al., 2023), Iterative DPO (Xu et al., 2023) and Self-Rewarding methods (Yuan et al., 2024). They can also be used for evaluation; e.g., many popular benchmark leaderboards are built by using a fixed LLM-as-a-Judge evaluation model (Li et al., 2023) such as GPT4 (Achiam et al., 2023).

We propose a novel recipe for training such an evaluator. Our overall method is an iterative training scheme that bootstraps improvements by annotating the current model's judgments using constructed synthetic data – so that the Self-Taught Evaluator is more performant on the next iteration.

Our overall pipeline is thus as follows:

- **Initialization:** We assume access to a large set of human-written user instructions, e.g., of the type that is commonly collected in production systems, and an initial seed LLM.

- **Instruction Selection:** We next select a challenging, balanced distribution of user instructions from the uncurated set by categorizing them via LLM.

- **Response Pair Construction:** For each user instruction (example) we create a preference pair of two model responses (chosen & rejected), generating them via prompting such that the rejected response is likely of lower quality than the chosen response.

- **Iterative Training:** We then iterate the following two steps:

  (i) **Judgment Annotation:** For each example, we sample from the current model up to $N$ times LLM-as-a-Judge generated reasoning traces and judgments. If we find a correct judgment we add that example to our training set, otherwise we discard it.

  (ii) **Model Fine-tuning:** We fine-tune the model on the newly constructed training set which yields an updated model for the next iteration.

Note that in each iteration of training the size of the training set depends on the quality of the current model. We expect that as the model improves, the size of the training set will increase as well, as the model will be able to find more correct judgments, giving the model a kind of automatic curriculum.

## 3.1 INITIALIZATION

We assume we have access to a pool of user instructions $\{x_i\}$. Each sample $x_i$ can either be one single text instruction or a multi-turn dialog history of turns between the user and the assistant, with the last turn being an instruction or question from the user. Instructions typically involve different skills such as general knowledge and reasoning, coding, safety, and mathematical reasoning.

## 3.2 INSTRUCTION SELECTION

Given a pool of human-written user instructions, there may be a large degree of noise, as well as an imbalance in terms of topic, variety, difficulty, and ability of the model to answer. We therefore aim to select a subset of instructions to generate high-quality synthetic responses and judgments that can be further used for training.

We classify each input using an LLM into a given category, for example coding, reasoning, brainstorming, etc. The precise prompt we use is given in Figure 7. We are then free to select data from within those categories, and to discard certain categories not deemed to be useful for training.

## 3.3 RESPONSE PAIR CONSTRUCTION

For each input $x_i$ in our curated training pool, we next generate preference data involving two responses $y_i^{(w)}$ and $y_i^{(l)}$ where $w$ is expected to be preferable (winning) over $l$ (losing). We achieve this by generating the data in a synthetic manner without using human annotation.

Given the instruction $x_i$, we first prompt an instruction-following LLM to generate a baseline response $y_i^w$ as usual. We then prompt the LLM to generate a "noisy" version of the original instruction $x_i' = \phi(x_i)$. We do this using the prompt template given in Figure 2, where we ask to "generate a modified instruction that is highly relevant but not semantically identical to the instruction above from the user." We then prompt the LLM for a high-quality response $y_i^l$ to $x_i'$, which would not be a good response for $x_i$. This yields a synthetic preference $y_i^w \succ y_i^l$ for the original input $x_i$.

This paired data is then used to construct training examples:

$$(x_i, y_i^{(A)}, y_i^{(B)})$$

where we randomize the order of whether the winner is $w = A$ or $w = B$, which is important to deal with position bias for LLM-as-a-Judge inference.

## 3.4 JUDGMENT ANNOTATION

Our LLM-as-a-Judge model is used to generate evaluation judgments (reasoning chains and verdicts) $\{j_i\}$ for each training example $e_i := (x_i, y_i^{(A)}, y_i^{(B)})$ in the following manner: for a given input $e_i$, we collect $N$ diverse evaluations $\mathcal{J} := \{j_i^1, \ldots, j_i^N\}$ by sampling from the model. We then apply rejection sampling to filter $\mathcal{J}$ by removing $j_i^n$ when the final verdict disagrees with the ground truth labeling, derived from Subsection 3.3. We then select a single correct reasoning chain and verdict at random from the pool of correct solutions. If no such judgment exists ($\mathcal{J}$ is empty) then we discard the example.

This now allows us to construct our final training examples of synthetic preferences for fine-tuning:

$$((x_i, y_i^{(A)}, y_i^{(B)}), j_i).$$

## 3.5 MODEL FINE-TUNING (ITERATIVE TRAINING)

Our Self-Taught Evaluator (LLM-as-a-Judge model) is first initialized with the seed LLM. The model is then trained in an iterative manner. At each iteration, we annotate the training examples with judgments as described in Subsection 3.4 using the current model, giving training examples $\{(x_i, y_i^{(A)}, y_i^{(B)}, j_i)\}$. These are used to train the next iteration's model by fine-tuning. Note that we initialize from the seed model at each iteration.

# 4 EXPERIMENTS

## 4.1 EXPERIMENTAL SETUP

**Training.** Our initial model $M_0$ is initialized from Llama3-70B-Instruct. In each iteration $i = 1, \ldots T$, we use model $M_{i-1}$ from the previous iteration to generate synthetic preferences followed by judgments on the training data, and then fine-tune Llama3-70B-Instruct again. We use fairseq2 library (Balioglu, 2023) for instruction finetuning and vLLM (Kwon et al., 2023) for inference. During training the negative log-likelihood loss is only applied to the evaluation part, i.e., $j_i$ of the training example. Training hyperparameters are provided in Table 7. Model selection is done using a combination of pairwise judgment accuracy and position bias computed over the held out set. Sampling parameters used for generations are provided in Table 8.

**Instructions and Responses.** We start with a large pool of human-written instructions $\{x_i\}$ from the WildChat dataset (Zhao et al., 2024). To perform prompt selection, we annotate the category of each instruction with the Mixtral 22Bx8 Instruct model, using the template in Figure 7 and select 20,582 examples in the reasoning category, as we expect these to be challenging inputs. For the selected inputs we generate synthetic responses $y_i^w$ and $y_i^l$ using Mixtral 22Bx8 Instruct following Subsection 3.3 and Figure 2.

**Judge Annotation.** For each training example, we sample $N = 15$ judgments from the model $M_{i-1}$ and retain one positive sample $j_i$ per example. Then over the entire dataset we sample the same amount of examples from different labels ("A is better", "B is better") to ensure balanced training. Judgements for training $M_0$ were sampled from Mixtral 22Bx8 Instruct, and from the Llama model being trained in all subsequent iterations.

The training data is constructed as ($<$system prompt$>$, $\{(x_i, y_i^{(A)}, y_i^{(B)}, j_i)\}$). We generate 10k synthetic examples for the first iteration of training. We use the standard system prompt from MT-Bench and RewardBench as shown in Figure 8.

**Majority Vote Inference.** As LLM-as-a-Judge uses chain-of-though reasoning chains generated by the LLM followed by a verdict, it is known that majority vote inference can yield improvements in these cases (Wang et al., 2023b). At inference time when evaluating final performance we sample generations $N$ times, and take the final judgment to be the most common verdict.

## 4.2 OTHER DATA SOURCES

To understand the effectiveness of the proposed method, we generate synthetic judgments using the same approach but based on the following data sources:

- HelpSteer2 (Wang et al., 2024c). We generate evaluations conditioned on the scores of helpfulness, correctness, coherence, complexity and verbosity provided the dataset. We use the aggregated score to derive the ground truth preference for each example using the recommended weighting $[0.65, 0.8, 0.45, 0.55, -0.4]$[1].
- GSM8K (Cobbe et al., 2021). We sample from an instruction-following model multiple times to get $y^w$ when the final solution agrees with the ground truth and $y^l$ vise versa.
- Coding instructions from WildChat. Similar to the "reasoning" prompts we selected from WildChat used in the main experiment, we also experimented with prompts annotated with the "Coding" category.
- hh_rlhf (Bai et al., 2022). We generate evaluations on the prompts and responses provided in the "harmless_base" training split. Then we take human preferences provided by the dataset as ground truth to perform rejection sampling to construct judgments.

## 4.3 EVALUATION

We evaluate the accuracy of our Self-Taught Evaluator model on the following benchmarks:

---

[1]Recommended weighting was taken from https://huggingface.co/nvidia/Llama3-70B-SteerLM-RM.

- RewardBench (Lambert et al., 2024). We use the standard evaluation protocol provided by the leaderboard.

- MT-Bench (Zheng et al., 2023). We report agreement rate with human judgments when examples with ties are excluded.

- HelpSteer2 (Wang et al., 2024c). We evaluate on the validation split.

## 5 RESULTS

### 5.1 REWARDBENCH

Results on RewardBench are given in Table 1. We find that our Self-Taught Evaluator which is trained iteratively on synthetic data *without* any annotated preference labels significantly improves over the seed Llama3-70B-Instruct model, matching top-performing reward models trained *with* labeled data. Our approach improves its results across training iterations, and achieves an overall score of 88.3 on iteration 5, while the seed model it starts from obtains 75.4. Training an LLM-as-a-Judge in a similar manner starting from the same seed using the labeled HelpSteer2 data we only obtain 85.6, hence we obtain superior performance *without using human labeled data*. Compared to the seed model, we observe improvements using our approach in evaluating instructions in the Chat Hard, Safety and Reasoning categories, while being worse on the easier Chat category – perhaps because our unlabeled training data focused the model on harder examples.

**Improving results further with majority voting** As also shown in Table 1, with 32-sample majority voting, our third iteration of Self-Taught Evaluator model reaches an overall performance of 88.7 on RewardBench, outperforming many other existing reward models.

### 5.2 MT-BENCH

We report results on MT-Bench in Table 2. Unlike RewardBench, the MT-Bench dataset contains tie votes (A and B are considered equally good). Since our models are trained to give binary decisions, we only report the agreement on non-tie examples. For each pair of responses A and B, we test two orders: where response A appears first and response B appears first, and average the results. We find that our Self-Taught Evaluator again outperforms the Llama3-70B-Instruct seed model, and is on par or slightly outperforms GPT4-0125.

### 5.3 HELPSTEER2

Results on the HelpSteer2 validation set are given in Table 3. We report the average accuracy of two orders and three seeds by swapping the response order in a similar manner, as well as reporting both orders separately (right answer first or second) to test for position bias. We further compute the position-consistent accuracy, treating a judgment as incorrect when a model has different predictions on the two orderings. We use the human labels from the Helpsteer2 dataset and treat the response with higher summed scores as the better response. We find that our Self-Taught Evaluator method improves both average accuracy and position-consistent accuracy compared to the seed Llama-3-70B-Instruct model.

## 6 ABLATIONS AND ANALYSIS

### 6.1 SYNTHETIC DATA FROM OTHER SOURCES

In Table 4, we compare Self-Taught Evaluator models trained on synthetic preferences constructed from different sources. We found data sources focusing on different skills, such as coding, mathematical reasoning, etc. are all effective in turning a strong instruction-following LLM into a strong LLM-as-a-Judge. Intuitively, we find that data sources generally improve the categories in Reward-Bench that are related to their distribution.

| Model | Overall | Chat | Chat Hard | Safety | Reasoning |
|---|---|---|---|---|---|
| Llama-3-70B-Instruct (seed) | 75.4 | 97.6 | 58.9 | 69.2 | 78.5 |
| *Self-Taught Evaluator, trained on synthetic data only* | | | | | |
| Iteration 1 | 83.9 | **98.3** | 69.0 | 85.7 | 82.6 |
| Iteration 2 | 86.0 | 97.5 | 75.4 | 89.5 | 81.7 |
| Iteration 3 | 87.5 | 97.2 | 79.1 | 89.7 | 83.9 |
| Iteration 4 | 87.7 | 98.0 | 80.3 | 90.5 | 82.2 |
| Iteration 5 | 88.3 | 96.6 | **84.2** | 91.5 | 81.0 |
| *w/ majority voting using 32 samples* | 88.7 | 96.9 | 84.0 | 91.5 | 82.5 |
| *Baselines with Labeled Data* | | | | | |
| Llama-3-70B-Instruct w/ HelpSteer2, LLM-as-a-Judge | 85.6 | 96.9 | 70.0 | 88.8 | 86.7 |
| nvidia/Llama3 70B RM with HelpSteer2, classifier * | **88.8** | 91.3 | 80.3 | **92.8** | 90.7 |
| *Other SoTA LLM-as-a-Judge baseline models* | | | | | |
| GPT4 0125 * | 84.3 | 95.3 | 74.3 | 87.2 | 86.9 |
| Gemini 1.5 Pro 0514 * | 88.1 | 92.3 | 80.6 | 87.5 | **92.0** |
| Llama3.1-405B-Instruct | 83.7 | 98.0 | 75.1 | 74.7 | 86.8 |
| Llama3.1-70B-Instruct | 82.2 | 97.8 | 69.7 | 76.3 | 85.2 |

Table 1: **RewardBench Results**. Our Self-Taught Evaluator trained on synthetic data without any human annotated preference labels matches top-performing reward models trained with labeled data. Models marked with (*) are taken from the RewardBench leaderboard.

| Model | Agreement with Human |
|---|---|
| Llama-3-70B-Instruct (seed) | 77.8 |
| *Self-Taught Evaluator, trained on synthetic data only* | |
| Iteration 1 | 79.0 |
| Iteration 2 | 78.7 |
| Iteration 3 | 78.9 |
| Iteration 4 | 77.5 |
| Iteration 5 | 78.9 |
| *w/ majority voting using 32 samples* | **79.5** |
| *Other SoTA LLM-as-a-Judge baseline models* | |
| GPT4-0125 | 79.1 |

Table 2: **MT-Bench Results**. Our Self-Taught Evaluator trained on synthetic data without any human annotated preference labels performs on par with GPT-4 judgments.

| Model | 0-1 Acc | 1-0 Acc | Avg Acc | Position-consistent Acc |
|---|---|---|---|---|
| Llama-3-70B-Instruct (seed) | 65.2 | 65.8 | 65.5 | 56.5 |
| *Self-Taught Evaluator, trained on synthetic data only* | | | | |
| Iteration 1 | 68.1 | 68.7 | 68.4 | 59.4 |
| Iteration 2 | 69.6 | 69.4 | 69.5 | 58.8 |
| Iteration 3 | 70.3 | 71.2 | 70.8 | 61.1 |
| Iteration 4 | 71.0 | 71.7 | **71.4** | **61.9** |
| Iteration 5 | 71.6 | 70.3 | 71.0 | 60.6 |

Table 3: **HelpSteer2 results**. Iterative training on synthetic preferences improves position-consistent accuracy compared to Llama3-70B-Instruct, measured on the HelpSteer2 (Wang et al., 2024c) validation split.

| Model | Source for synthetic preferences | Overall | Chat | Chat Hard | Safety | Reasoning |
|---|---|---|---|---|---|---|
| Llama-3-70B-Instruct | | 75.4 | **97.6** | 58.9 | 69.2 | 78.5 |
| | safety (hh_rlhf) | 79.6 | 97.2 | 55.4 | **87.0** | 78.8 |
| | math (GSM8K) | 79.3 | 96.1 | 58.8 | 79.4 | **83.0** |
| | coding (WildChat) | 79.4 | 96.6 | 55.9 | 85.3 | 79.7 |
| | reasoning (WildChat) | 83.5 | 97.5 | **70.6** | 84.2 | 81.6 |

Table 4: Supervised fine-tuning with synthetic preferences from different sources improves Llama-3-70B-Instruct on various categories, as measured on RewardBench. Largest improvement in each category is highlighted in bold.

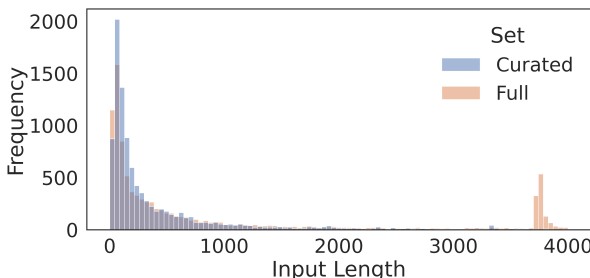

Figure 3: Distribution of curated training set of selected instructions compared to the full WildChat dataset.

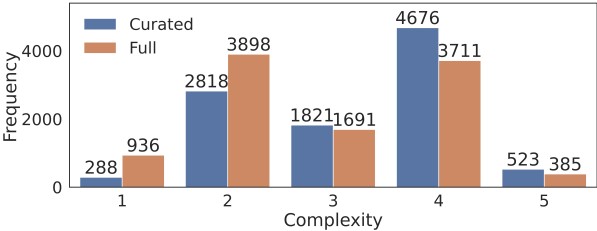

Figure 4: Distribution of inferred complexities of curated training data versus all instructions in WildChat.

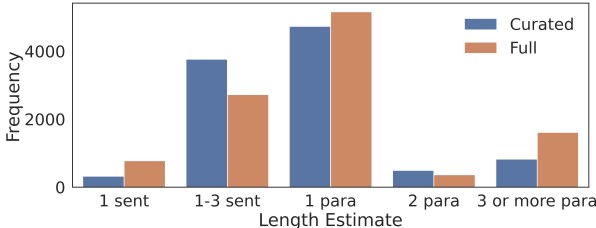

Figure 5: Distribution of estimated output lengths of curated training data versus all instructions in WildChat.

## 6.2 SYNTHETIC BAD RESPONSE GENERATION

In our experiments we generate synthetic data by first generating a modified instruction, and then a good response for the modified instruction – with the aim that this will be a bad response for the original instruction. An alternative is to just prompt an LLM to generate a bad response to the original instruction directly. We use the prompt template given in Figure 10 and otherwise conduct training as before on the same set of reasoning-based instructions. This approach obtains a RewardBench overall score of 80.7, which still works – but is worse than using our proposed approach, which achieves 83.8.

## 6.3 COMPARISON OF SYNTHETIC DATA WITH HUMAN ANNOTATED DATA

We conducted the same iterative training using labeled preference data from HelpSteer2 (Wang et al., 2024c), rather than synthetic data. On RewardBench, as is shown in Table 5, the improvement from each iteration is smaller and the final model did not outperform iterative training on synthetic preferences. We note that these experiments use data to train an LLM-as-a-Judge. Other results in the literature have used the HelpSteer2 to train classifier-based reward models with slightly better results on RewardBench, e.g., obtaining 88.8 using Llama-3-70B, see Table 1.

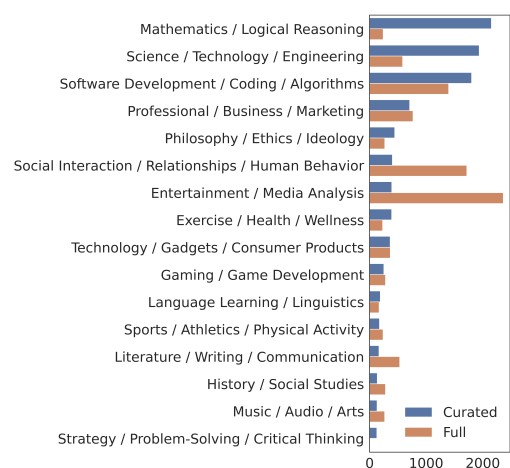

Figure 6: Distribution of inferred categories of curated training data versus all instructions in Wild-Chat.

| Model | Overall | Chat | Chat Hard | Safety | Reasoning |
|---|---|---|---|---|---|
| Llama-3-70B-Instruct (seed) | 75.4 | **97.6** | 58.9 | 69.2 | 78.5 |
| *Self-Taught Evaluator, trained on labeled HelpSteer2 preferences* | | | | | |
| Iteration 1 | 85.6 | 96.9 | 70.0 | 88.8 | 86.7 |
| Iteration 2 | 86.3 | 96.1 | 72.4 | 91.1 | 85.7 |
| Iteration 3 | **87.0** | 95.0 | 74.2 | 91.2 | **87.8** |
| Iteration 4 | **87.0** | 94.1 | **77.2** | **91.6** | 85.1 |

Table 5: Iterative training with labeled data also shows improvement on RewardBench. However, it does not outperform iterative training with synthetic preferences .

| synthetic:HelpSteer2 ratio | Overall | Chat | Chat Hard | Safety | Reasoning |
|---|---|---|---|---|---|
| 1 : 0 | 0.835 | **0.975** | 0.706 | 0.842 | 0.816 |
| 0 : 1 | 0.856 | 0.969 | 0.700 | 0.888 | **0.867** |
| 1 : 1 | 0.842 | 0.972 | 0.681 | 0.881 | 0.836 |
| 1 : 2 | **0.858** | 0.972 | 0.711 | **0.891** | 0.857 |
| 1 : 5 | 0.847 | **0.975** | 0.681 | 0.889 | 0.844 |
| 2 : 1 | 0.833 | 0.972 | 0.689 | 0.847 | 0.823 |
| 5 : 1 | **0.858** | 0.972 | **0.726** | 0.880 | 0.853 |

Table 6: Mixing data sources in different proportions can improve performance of the fine-tuned model. Synthetic preference data is generated with the Llama3-70B-Instruct model.

## 6.4 ITERATIVE TRAINING BY INITIALIZING FROM LABELED DATA

We further explore how to utilize labeled data in our pipeline. We first finetune a model on Help-steer2 Wang et al. (2024c) and use this model to generate judgements. In this way, we obtain synthetic data by utilizing a model finetuned on labeled data. We conducted iterative training and present results in Table 12. We observed good performance compared to the seed model (Llama-3-70B-Instruct), however it does not clearly outperform conducting iterative training with unlabeled data alone.

## 6.5 COMBINING SYNTHETIC AND HUMAN LABELED PREFERENCE DATA

We compare how combining synthetic preference data with human labelled preference data affects model performance. In particular, we combine synthetic preferences generated from reasoning Wild-Chat prompts with the human labeled HelpSteer2 dataset (train split) and report performance in Table 6. We compare to first-iteration models trained on single data source, and select the best checkpoint for joint training using the validation split of HelpSteer2 and holdout set of synthetic preferences (in-distribution), as well as safety and code synthetic preferences (out-of-distribution).

We then report evaluation results on RewardBench. The results show that overall the models retain strong performance across different data mixing weights, with slight improvements on overall accuracy.

## 6.6 INSTRUCTION COMPLEXITY

We analyze the length distribution of the curated training set of selected instructions in Figure 3. The dataset has a long-tail distribution of input length, with most of the examples less than 500 tokens. In contrast, the full dataset (i.e., the full data before the instruction selection step of Subsection 3.2) has a cluster of very long instructions, containing content such as long-form coding instructions or transcripts.

We further instruct Llama-3-70B-Instruct to infer the complexity (using a score of 1–5) and category of each input instruction, as well as the length of the expected output, following the procedure in Yuan et al. (2024). From Figure 4 and Figure 6, we see that the curated dataset has more complex instructions involving logical reasoning/science whereas the full dataset has a greater proportion focused on relationships and entertainment. Finally, in Figure 5 we see that the anticipated length of the response is higher for the full dataset than the curated one, perhaps because of the greater frequency of lengthy, and sometimes repetitive instructions.

## 7 CONCLUSION

We present a scalable approach to build a strong generalist evaluator to perform model-based evaluation of LLM outputs. Our method constructs synthetic preferences over pairs of responses without using any human annotation. Our Self-Taught evaluator with iterative training over these synthetic preferences greatly boosts the accuracy of a strong seed LLM (Llama3-70B-Instruct) as an evaluator, from 75.4 to 88.7 on RewardBench, a new state-of-the-art for generative LLM-as-a-Judge methods.

## 8 LIMITATIONS

Generative LLM-as-a-Judge models usually have longer outputs and thus higher inference cost than reward models that simply output a score, as LLM-as-a-Judge typically first generates a reasoning chain. On the other hand, models that generate long reasoning chains are more susceptible to producing hallucinated content. This highlights a trade-off between encouraging deeper reasoning and mitigating the risk of generating inaccurate or fabricated information. Further, we have used relatively large LLMs in this work (70B parameters) and made no study of whether our approach works on smaller models. Since we use a seed model to generate first synthetic preferences during our iterative training scheme, one of the assumptions is that the model is capable of generating reasonable evaluations. Thus, our approach is limited by having a capable instruction fine-tuned model which is already reasonably aligned to human (or legal/policy) preferences. Furthermore, we only investigated and reported metrics involving evaluation accuracy improvements, rather than computational requirement concerns. While LLM-as-a-judge models can also be utilized to provide reward signals for optimizing LLM performance, our evaluation did not explore this application. Future work could investigate the potential benefits of using our model in this context. We also only investigated *pairwise evaluation*, i.e., comparing two responses, whereas it is also possible to use LLM-as-a-Judge models (or any other model) to evaluate the quality of *single responses*, e.g., giving them a score out of 5 or 10, rather than a pairwise A vs B judgment. We leave evaluating single responses to future work.

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

## A APPENDIX

### A.1 PROMPT TEMPLATES

We provide the prompt templates used for annotating and selecting instructions (Figure 7), annotating judgments with synthetic preferences (Figure 8), and generating ablation synthetic preference data with bad responses (Figure 10). Figure 9 illustrates an training example based on synthetic preference data.

### A.2 MORE TRAINING AND EVALUATION DETAILS

We include training hyper-parameters in Table 7 and sampling parameters in Table 8.

| Name | Value |
|---|---|
| max_seq_len | 4096 |
| max_num_tokens | 8192 |
| model | llama3_70b_instruct |
| dtype | bfloat16 |
| data_parallelism | fsdp |
| tensor_parallel_size | 8 |
| activation_checkpointing | true |
| lr | 1.0e-06 |
| betas | 0.9, 0.95 |
| final_lr_ratio | 0.2 |
| weight_decay | 0.1 |
| num_lr_warmup_steps | 100 |
| gradient_accumulation | 1 |
| max_num_data_epochs | 2 |
| checkpoint_every_n_steps | 100 |
| seed | 2 |

Table 7: Training hyper-parameters used during fine-tuning.

---

**Prompt Template for Selecting Instructions**

I have an instruction below that I would like you to perform three steps of analysis about the instruction:

<instruction> {instruction} </instruction>

Firstly, categorize the instruction above into one of the following categories:

Coding
Mathematical reasoning
Asking for Advice
Brainstorming
Classification
Closed Question Answering
Creative Writing
Extraction
Inhabiting a Character/Persona
Open Question Answering
Rewriting
Summarization
Knowledge and Reasoning
Humanity, History or Social Studies
Other

Secondly, score the instruction in terms of complexity: how complex you think it is to answer from 1-10 (where 10 is a complex question whereby first reasoning or breaking down the question into multiple subquestions for example might help improve the answer).

Thirdly, indicate how long you think the response to the instruction should be, either (a) 1 sentence, (b) 1-3 sentences, (c) 1 paragraph, (d) 2 paragraphs, or (e) 3 or more paragraphs.

Provide your final response in the following format:
Category: <one of the categories above>
Complexity: <score out of 10>
Length: <choose from (a) to (e)>. DO NOT provide the actual response.

---

Figure 7: **Prompt template for Selecting Instructions.** We prompt an instruction following model to annotate the category of each instruction in order to curate our training data instructions.

| Stage | Generation for | Temperature | Top p |
|-------|----------------|-------------|-------|
| Train | Judgment | 0.7 | 0.9 |
| Eval | MT-Bench | 0.0 | 1.0 |
| Eval | Reward Bench (RB) | 0.0 | 1.0 |
| Eval | RB w/ maj voting | 0.7 | 0.9 |
| Eval | Helpsteer 2 valid | 0.7 | 0.9 |

Table 8: Sampling parameters (temperature and top p) used during generations at each stage of training and evaluation.

### A.3 POSITION ORDER EVALUATION ON REWARDBENCH

We notice that when we evaluate generative models on RewardBench, the order of two responses in each example is not fixed. More specifically, for each example, the winning response ($y^w$) can be randomly placed before or after the losing response ($y^l$). Generative models may output different judgements when the order of responses changes. Thus, we analyze how the performance varies when different seeds are used to decide response order. In Table 9, we test our model from the 5th iteration of training on RewardBench with the response order randomly shuffled, as well as two extreme cases where the winning answer always appear first or last. We recommend to report the average performance (88.3 for our 5th iteration) of "$y^w$ always first" and "$y^l$ always first" as it fairly considers both orders.

> **Prompt Template for Judgment Annotation**
>
> Please act as an impartial judge and evaluate the quality of the responses provided by two AI assistants to the user question displayed below. You should choose the assistant that follows the user's instructions and answers the user's question better. Your evaluation should consider factors such as the helpfulness, relevance, accuracy, depth, creativity, and level of detail of their responses. Begin your evaluation by comparing the two responses and provide a short explanation. Avoid any position biases and ensure that the order in which the responses were presented does not influence your decision. Do not allow the length of the responses to influence your evaluation. Do not favor certain names of the assistants. Be as objective as possible. After providing your explanation, output your final verdict by strictly following this format: "[[A]]" if assistant A is better, "[[B]]" if assistant B is better.
>
> Please act as an impartial judge and evaluate the quality of the responses provided by two AI assistants to the user question displayed below. You should choose the assistant that follows the user's instructions and answers the user's question better. Begin your evaluation by first verifying whether each response contains any obvious or subtle errors. Then propose an appropriate evaluaiton rubric, e.g. 1-5 criteria that are important for evaluating responses to this specific user question. Continue your evaluation by checking each response carefully along those criteria. Based on the analysis in previous steps, choose which response is better overall. Avoid any position biases and ensure that the order in which the responses were presented does not influence your decision. Do not allow the length of the responses to influence your evaluation. Do not favor certain names of the assistants. Be as objective as possible. After providing your evaluation, output your final verdict by strictly following this format: "[[A]]" if assistant A is better, "[[B]]" if assistant B is better.
>
> [[User Question]]
> {instruction}
>
> [The Start of Assistant A's Answer]
> {response A}
> [The End of Assistant A's Answer]
>
> [The Start of Assistant B's Answer]
> {response B}
> [The End of Assistant B's Answer]

Figure 8: **Prompt template for Judgment Annotation.** This is the same prompt as used in MT-Bench and RewardBench.

| Seed | Average Accuracy |
|---|---|
| 1 | 88.9 |
| 11 | 88.4 |
| 111 | 88.6 |
| 1111 | 88.7 |
| 11111 | 88.3 |
| $y^w$ always first | 85.5 |
| $y^l$ always first | 91.1 |

Table 9: Average accuracy on RewardBench when order of responses changes.

## A.4 USING DIFFERENT MODELS FOR TRAINING DATA GENERATION

In Table 10 we present evaluation on RewardBench of models finetuned on different training data. Note in our Self-Taught Evaluator approach we can use different LLMs to generate responses and judgements. Specifically, we try using Mixtral 22Bx8 Instruct or Llama-3-70B-Instruct in various combinations. We then finetune the Llama-3-70B-Instruct model and test on RewardBench. As shown in Table 10, the model finetuned on data generated by using the Mixtral 22Bx8 Instruct model to judge Mixtral 22Bx8 Instruct model generated responses achieves the best performance.

| Model | Overall | Chat | Chat Hard | Safety | Reasoning |
|---|---|---|---|---|---|
| Llama-3-70B-Instruct (seed) | 75.4 | **97.6** | 58.9 | 69.2 | 78.5 |
| *Self-Taught Evaluator, trained on synthetic data only* | | | | | |
| Mixtral judge Mixtral | 83.9 | 98.3 | 69.0 | 85.7 | 82.6 |
| Llama3.0 judge Llama3.0 | 81.4 | 97.2 | 66.0 | 85.0 | 77.5 |
| Llama3.0 judge Mixtral | 80.0 | 97.5 | 70.0 | 72.8 | 79.4 |

Table 10: Performance on RewardBench of models finetuned on different training data.

| Model | Overall | writing | stem | coding | math | humanities | reasoning | roleplay | extraction |
|---|---|---|---|---|---|---|---|---|---|
| Llama-3-70B-Instruct (seed) | 77.8 | 70 | 76.9 | 73.8 | 80 | 79.85 | 78.8 | 78.8 | **85.1** |
| *Self-Taught Evaluator, trained on synthetic data only* | | | | | | | | | |
| Iteration 1 | 78.95 | **71.15** | 78.95 | 76.6 | 81.65 | 80.25 | 82.3 | **80.7** | 80.25 |
| Iteration 2 | 78.65 | 69.6 | 77.9 | 82.55 | 79.15 | 82.2 | 80.8 | 77.6 | 79.8 |
| Iteration 3 | 78.9 | 70.4 | 78.6 | 79.35 | 79.55 | 82.9 | 82.3 | 77.9 | 80.7 |
| Iteration 4 | 77.45 | **71.15** | 77.25 | 75.8 | 73.3 | 82.6 | 81.3 | 78.85 | 79.35 |
| Iteration 5 | 78.9 | 68.45 | 78.2 | 81.75 | 82.5 | 82.25 | 81.35 | 75.15 | 83.75 |
| *w/ majority voting @ 32* | **79.45** | 68.45 | 78.55 | **82.95** | **83.75** | 82.9 | **82.8** | 76.35 | 81.6 |
| *Other SoTA LLM-as-a-Judge baseline models* | | | | | | | | | |
| GPT4-0125 | 79.15 | 70.4 | **79.9** | 82.9 | 82.1 | 80.55 | 80.8 | 77 | 80.7 |

Table 11: **MT-Bench Per-category Results**. Our Self-Taught Evaluator trained on synthetic data without any human annotated preference labels performs on par with GPT-4 judgments.

| Model | Overall | Chat | Chat Hard | Safety | Reasoning |
|---|---|---|---|---|---|
| Llama-3-70B-Instruct (seed) | 75.4 | **97.6** | 58.9 | 69.2 | 78.5 |
| *Self-Taught Evaluator, trained on synthetic data generated by a finetuned model (Helpsteer2)* | | | | | |
| Iteration 1 | 87.0 | 95.8 | 75.8 | 90.7 | 85.8 |
| Iteration 2 | 86.6 | 92.2 | 77.4 | 91.2 | 85.8 |

Table 12: Iterative training on synthetic data generated by a model that is first fine-tuned on labeled data (Helpsteer2).

| Model | Overall | Chat | Chat Hard | Safety | Reasoning |
|---|---|---|---|---|---|
| Llama-2-70B-Instruct | 60.7 | 87.8 | 42.0 | 60.6 | 52.5 |
| Llama-2-70B-Instruct SFT iter1 | 74.4 | 96.6 | 53.9 | 76.7 | 70.4 |
| Llama-3-8B-Instruct | 64.4 | 85.5 | 41.6 | 67.5 | 64.8 |
| Llama-3-8B-Instruct SFT iter1 | 71.5 | 92.7 | 44.5 | 78.6 | 70.1 |
| Llama-3-70B-Instruct | 75.4 | 97.6 | 58.9 | 69.2 | 78.5 |
| Llama-3-70B-Instruct SFT iter1 | 83.9 | 98.3 | 69.0 | 85.7 | 82.6 |
| Llama-3.1-70B-Instruct | 82.8 | 97.5 | 71.3 | 76.9 | 85.4 |
| Llama-3.1-70B-Instruct SFT iter1 | 86.2 | 96.1 | 76.1 | 86.8 | 85.7 |

Table 13: We applied our Self-taught evaluator approach to the LLaMA2, LLaMA3, and LLaMA3.1 models. We present results after the first iteration of supervised fine-tuning. Our approach consistently demonstrates performance improvement across different models, even with just one iteration.

| Model | Overall | Chat | Chat Hard | Safety | Reasoning |
|---|---|---|---|---|---|
| Skywork-Critic-Llama-3.1-70B (Shiwen et al., 2024) | 93.3 | 96.6 | 87.9 | 93.1 | 95.5 |
| SFR-LLaMa-3.1-70B-Judge-r (Wang et al., 2024b) | 92.7 | 96.9 | 84.8 | 91.6 | 97.6 |
| Self-Taught Evaluator | 88.3 | 96.6 | 84.2 | 91.5 | 81.0 |
| flame-1.0-24B-july-2024 (Vu et al., 2024) | 87.8 | 92.2 | 75.7 | 89.6 | 93.8 |
| CompassJudger-1-32B-Instruct (Cao et al., 2024) | 85.2 | 98.0 | 65.1 | 85.3 | 92.4 |
| prometheus-8x7b-v2.0 (Kim et al., 2024) | 74.5 | 93.0 | 47.1 | 80.5 | 77.4 |
| prometheus-7b-v2.0 (Kim et al., 2024) | 72.0 | 85.5 | 49.1 | 77.1 | 76.5 |

Table 14: We present a comparison between our Self-taught evaluator and several other LLM-as-a-judge models. The state-of-the-art (SOTA) performance is achieved by Shiwen et al. (2024), where they fine-tune the Llama-3.1-70B instruct model on a pool of various human-labeled preference datasets, totaling 80K pairs. The remaining models (except for our Self-taught evaluator) are built on top of different base models but all rely on human-labeled preference datasets. In contrast, our Self-taught evaluator is based on the Llama-3.0-70B instruct model and only 10K synthetic pairs. Despite this, it still achieves good performance, demonstrating its effectiveness as an evaluator.

| Original Question and Response | Modified Question and Response |
|---|---|
| **Question** ($x$): if detroit is the center of automotive industry of USA, then why it is so poor?
**Response** ($y^w$): Detroit, once known as the "Motor City," was the heart of the American automotive industry for many years. However, several factors have contributed to its diminished role in this sector:
1. Globalization and competition: The rise of foreign automakers and the globalization of the automotive industry have posed significant challenges to Detroit's automakers. Companies like Toyota, Honda, and Volkswagen have gained market share, forcing Detroit's Big Three (General Motors, Ford, and Chrysler) to adapt and compete in a more global market.
2. Technological advancements: The advent of new technologies, such as electric and autonomous vehicles, has disrupted the traditional automotive industry. Detroit's automakers have had to invest heavily in research and development to stay competitive, which has put pressure on their financial performance.
3. Labor issues: The United Auto Workers (UAW) union has a strong presence in Detroit, and labor disputes and strikes have occasionally disrupted production for the Big Three. This has led to increased labor costs and reduced competitiveness compared to non-unionized foreign automakers.
4. Environmental regulations: As environmental concerns have grown, regulatory pressures on the automotive industry have increased. Detroit's automakers have had to invest in cleaner technologies and comply with stricter emissions standards, which has added to their financial burden.
5. Industry consolidation: Over the years, there has been consolidation within the automotive industry, with mergers and acquisitions reshaping the competitive landscape. This has affected Detroit's automakers, as they have had to adapt to new ownership structures and strategic priorities.
These factors have collectively contributed to the decline of Detroit's prominence in the automotive industry. However, the city's automakers are working to adapt and innovate in response to these challenges. | **Question**($x'$): What factors contribute to the economic disparity in Detroit, despite its historical significance as a hub for the US automotive industry?
**Response** ($y^l$): The economic disparity in Detroit can be attributed to several factors, including:
1.Decline of the automotive industry**: Mass layoffs and plant closures led to a significant loss of jobs and tax revenue.
2.Suburbanization and white flight**: Middle-class residents moved out of the city, taking their tax base with them, and leaving behind a concentrated population of low-income residents.
3.Lack of diversification**: Detroit's economy was heavily reliant on the automotive industry, making it vulnerable to fluctuations in the market.
4.Racial and socioeconomic disparities**: Historical discriminatory practices, such as redlining and segregation, have contributed to persistent racial and socioeconomic disparities in the city.
5.Government policies and corruption**: Poor governance, corruption, and inefficient use of resources have hindered the city's ability to address its economic challenges.
These factors have cumulatively contributed to the economic disparity in Detroit, despite its historical significance as a hub for the US automotive industry. |
| **Question** ($x$): There are 3 objects arranged in a line. Someone says "the table is to the right of the bag, the table is to the left of the box and the bag is to the right of the box." Is the situation just described possible or impossible? Before answering, take a deep breath. Think step by step before giving a final answer. My survival depends on your answer being correct, so concentrate hard.
**Response** ($y^w$): Let's break down the statements:
"The table is to the right of the bag."
"The table is to the left of the box."
"The bag is to the right of the box."
From statement 3, we know the order is: box $->$ bag
From statement 1, we know the order is: bag $->$ table
Combining these two, we get: box $->$ bag $->$ table
Now, let's look at statement 2: "The table is to the left of the box." This statement contradicts our previous conclusion.
If the table is to the left of the box, then the correct order should be: table $->$ box $->$ bag
However, this contradicts statement 1, which says the table is to the right of the bag.
Therefore, I conclude that the situation described is IMPOSSIBLE. The statements are mutually contradictory, and it's not possible for all three statements to be true at the same time. There are 3 objects arranged in a line. Someone says "the table is to the right of the bag, the box is to the left of the table and the bag is to the left of the box." Please analyze this new situation and determine whether it is possible or impossible. | **Question** ($x'$): There are 3 objects arranged in a line. Someone says "the table is to the right of the bag, the box is to the left of the table and the bag is to the left of the box." Please analyze this new situation and determine whether it is possible or impossible.
**Response** ($y^l$): Let's break down the given statements:
The table is to the right of the bag.
The box is to the left of the table.
The bag is to the left of the box.
From statement 3, we know the order is: bag $->$ box
From statement 2, we know the order is: box $->$ table
Now, let's combine these orders:
bag $->$ box $->$ table (from statements 2 and 3)
Statement 1 says the table is to the right of the bag, which is consistent with this order.
Based on the analysis, it is POSSIBLE for all three statements to be true at the same time. The correct order is: bag $->$ box $->$ table. |

Table 15: Examples of original question ($x$) and response ($y^w$) pair, as well as the modified question ($x'$) and the corresponding response ($y^l$).

---

**Example Judgment Generation**

**Instruction:**
`difference in percentage between 16200 and 12000`
**Response A:**
`The percentage difference between 16200 and 12000 is approximately 35%.`
`To calculate the percentage difference, you can use the following formula:`
`Percentage Difference = [(New Value - Original Value) / Original Value] x 100`
`In this case, the original value is 12000 and the new value is 16200.  So the`
`calculation would look like this:`
`Percentage Difference = [(16200 - 12000) / 12000] x 100`
`Percentage Difference = [4200 / 12000] x 100`
`Percentage Difference = 0.35 x 100`
`Percentage Difference = 35%`
`So the percentage difference between 16200 and 12000 is 35%.`
**Response B:**
`The percentage increase from 12000 to 16200 is approximately 35%.`
`To calculate the percentage change, you can use the following formula:`
`Percentage Calculation = [(New Value - Original Value) / Original Value] x 100`
`In this case, the original value is 12000 and the new value is 16200.  So the`
`calculation would look like this:`
`Percentage Calculation = [(16200 - 12000) / 12000] x 100`
`Percentage Calculation = [4200 / 12000] x 100`
`Percentage Calculation = 0.35 x 100`
`Percentage Calculation = 35%`
`So the percentage increase from 12000 to 16200 is 35%.`
**Judgment Generation:**
After carefully evaluating the responses from both Assistant A and Assistant B, I will provide my impartial judgment.
**Error Check:**
Both responses appear to be error-free in terms of calculation and mathematical operations.
**Evaluation Rubric:**
To evaluate the responses, I will consider the following 5 criteria:
1. **Accuracy**: Is the calculation correct, and is the answer accurate?
2. **Relevance** : Does the response directly address the user's question?
3. **Clarity**: Is the explanation clear, concise, and easy to understand?
4. **Formula Application**: Is the formula correctly applied, and are the steps clearly explained?
5. **Contextual Understanding**: Does the response demonstrate an understanding of the context and the user's intent?
**Evaluation:**
**Assistant A:**
* Accuracy: 5/5 (correct calculation)
* Relevance: 5/5 (directly addresses the user's question)
* Clarity: 5/5 (clear and concise explanation)
* Formula Application: 5/5 (correctly applied formula with clear steps)
* Contextual Understanding: 4/5 (uses "percentage difference" which is correct, but could be more explicit)
**Assistant B:**
* Accuracy: 5/5 (correct calculation)
* Relevance: 5/5 (directly addresses the user's question)
* Clarity: 5/5 (clear and concise explanation)
* Formula Application: 5/5 (correctly applied formula with clear steps)
* Contextual Understanding: 3/5 (uses "percentage calculation" which is correct in this case, but is less precise)
**Final Verdict:**
Based on the evaluation, I conclude that **[[A]]** is the better response.  Although both responses are accurate and clear, Assistant A demonstrates a slightly better understanding of the context and uses the more specific term "percentage difference", which is more applicable to this scenario.

---

Figure 9: An illustrative example of judgment generation given an instruction and two responses.

---

**Prompt Template for Generating a Bad Response for an Instruction**

Below is a conversation between an user and an AI Assistant.

[User Question]
{Instruction}

[The start of Assistant's Answer]
{Baseline Response}
[The end of Assistant's Answer]

Please rewrite the Assistant's Answer to make it worse. Specifically, the rewritten worse answer should closely resemble the original answer but is worse in terms of one or multiple of the following aspects: helpfulness, correctness, coherence, verbosity.
IMPORTANT: Please strictly follow the following format:
First, choose one or multiple aspects to generate a worse answer, such as rewrite the original answer to be unhelpful, incorrect, lack of coherence, more verbose, etc.
[The start of a rewritten worse answer]
<provide a worse answer here>
[The end of a rewritten worse answer]

---

Figure 10: **Generating a Bad Response for an Instruction.** This approach is an ablation compared to our proposed approach described in the main paper.

