# OpenReview forum: "Self-Taught Evaluators"
_ICLR.cc/2025/Conference — Submitted to ICLR 2025_

### Official Review · Reviewer_EbF7 · 2024-10-28

**Soundness:** 3
**Presentation:** 2
**Contribution:** 3
**Rating:** 6
**Confidence:** 5

**Summary:**

This paper introduces a novel approach focused on leveraging synthetic training data to enhance evaluators without the need for resource-intensive human annotations during model development. Moreover, it delves into an iterative self-training methodology and introduces an iterative self-improvement framework capable of producing contrasting outputs and training an LLM-as-a-Judge. The proposed Self-Taught Evaluator demonstrates notable performance enhancements on RewardBench.

**Strengths:**

(1) Writing: This paper is well-written and easy-to-understand.

(2) Method: Instead of adhering to the conventional LLM-as-a-Judge paradigm for providing judgment explanations, this paper introduces an innovative approach leveraging synthetic training data to enhance evaluators, thus avoiding the need for expensive and time-consuming human annotations during model development.

(3) Experiments: Through iterative experiments on LLama3-70B-Instruct, this paper convincingly showcases the effectiveness of the proposed Self-Taught Evaluators across RewardBench, MT-Bench, and HelpSteer2 datasets.

**Weaknesses:**

(1) Concerning the explanation provided in line 044 and Figure 2, could you elaborate on the methodology used to ascertain the contrastiveness of the generated synthetic preference pairs beyond the prompt instructions?

(2) The definition of "similar" in Figure 2 appears ambiguous. Specifically, while x' is deemed similar to x, the term "similar" implies a high level of relevance without strict semantic identity. Clarification on this aspect would be beneficial.

(3) Several typographical errors, such as the presence of "?" in Figure 2 and line 241, have been noted.

(4) The citation format lacks consistency, as evidenced by variations between line 137 and line 139. It would be beneficial to standardize this aspect.

(5) In terms of experiments, aside from Llama, have results been obtained using different models of varying sizes?

(6) Could you provide insight into the interpretation of performance fluctuations observed in Chat and Reasoning across multiple iterations?

**Questions:**

Please see the weakness part.

---

> ### Author Response · Authors · 2024-11-28
> **Thank you for your comments.**
>
> > Concerning the explanation provided in line 044 and Figure 2, could you elaborate on the methodology used to ascertain the contrastiveness of the generated synthetic preference pairs beyond the prompt instructions?
>
> We extract the final verdict from the synthetic preference pairs to ensure they are contrastive pairs, i.e., the pair must have opposite final verdicts [[A]] and [[B]].
>
> > The definition of "similar" in Figure 2 appears ambiguous. Specifically, while x' is deemed similar to x, the term "similar" implies a high level of relevance without strict semantic identity. Clarification on this aspect would be beneficial.
>
> We have updated the paper to improve the accuracy of our writing. Specifically, we now refer to x' as the "modified instruction" instead of a "similar instruction".
>
> > Several typographical errors, such as the presence of "?" in Figure 2 and line 241, have been noted.
>
> Fixed.Thanks!
>
> > The citation format lacks consistency, as evidenced by variations between line 137 and line 139. It would be beneficial to standardize this aspect.
>
> Thank you for bringing this to our attention. We have noted that other reviewers share the same concern. In response, we have thoroughly proofread our paper to eliminate such inconsistencies and ensure consistency throughout.
>
> > In terms of experiments, aside from Llama, have results been obtained using different models of varying sizes?
>
> Please refer to Table 12 in the Appendix. We conducted experiments with Llama2-70B, Llama 3 (8B and 70B), and Llama 3.1 70B using our approach, limited to the first iteration. The results consistently show significant performance improvements across all models.
>
> > Could you provide insight into the interpretation of performance fluctuations observed in Chat and Reasoning across multiple iterations?
>
> Our experiments with RewardBench revealed a competition between different categories. For instance, across iterations, we observed that the performance of the "chat" category improved while the performance of the "chat-hard" category declined. We speculate that this is due to the distinct capabilities required by each category; the "chat" category emphasizes stylistic and writing tone aspects, whereas the "chat-hard" category prioritizes factuality and correctness. We believe that such competitions may contribute to performance fluctuations in certain scenarios.

---

### Official Review · Reviewer_ubyr · 2024-10-29

**Soundness:** 3
**Presentation:** 4
**Contribution:** 3
**Rating:** 5
**Confidence:** 3

**Summary:**

The authors propose a new approach to training an LLM-as-a-Judge that, unlike prior work, does not need human supervision.
It can improve the judgement quality of strong instruction-tuned LLMs to be comparable to SotA judge models trained in a supervised manner.

The method utilises clever prompting to generate pairs of completions for an instruction where one is known to be preferred to the other.
Judgements with chain-of-thought reasoning are then generated from the target LLM, with correct ones being SFT trained on in an iterative procedure.

It is tested on a variety of benchmarks, along with some ablations.
The original claim is somewhat supported by the empirical evidence.

**Strengths:**

The problem the paper is trying to address is somewhat well motivated.

The paper is well presented, with minimal errors.

The method is reasonably well communicated, and to this reviewers knowledge, reasonably novel, extending existing work nicely.
The experimental setup is also clearly communicated, with details on hyper-parameters that would greatly aid reproducibility.

The method is tested on a variety of datasets, and using a variety of data sources.
Additionally, scores per iteration and for different subsets of the RewardBench dataset are given.
Thus, the empirical data generated to evaluate this method is extensive.

Some of the empirical evidence supports the claim that the proposed method outperforms existing SotA reward models.

The authors perform several ablations and additional analysis of their method, empirically justifying some of their design choices.

**Weaknesses:**

It's not entirely clear why the method should work, especially if given many iterations.
See "Questions" section for more details.

Experimentally, it appears that the iterated training does not always help, with the score often decreasing or noisily bouncing around after the first iteration.
See Table 1 "Chat" and "Reasoning" columns, table 2, and table 5 "Chat" column for clear examples of this.
In many other columns, score does not monotonically increase with iteration.
Thus, it's not clear to what extent the iterative nature of this method provides *consistent* improvements.

From table 1, the method is beaten both overall and in the "Safety" and "Reasoning" categories by other methods.
This slightly undermines the claim that the method outperforms or matches existing top-performing reward models.

The main motivation behind training a reward model is to provide a reward signal to optimise a downstream LLM's outputs to better reflect human values.
The proposed method has not been evaluated on its ability to provide such a reward signal.

## Errata
* Figure 1 is not referenced in the main paper text
* Table 5 "Chat" column has the Iteration 3 element in bold, but the Iteration 1 element is highest scoring.

**Questions:**

The proposed method iterates over instruction and response pairs that are generated at the start, and then fixed.
Could this not cause the model to over-fit to these specific instructions and responses, neglecting performance on others?

Since the model is being fine-tuned to re-produce its own previous outputs (with some filtering applied), how can it learn to generate better critiques?
Where is the exploration coming from, and why would it favor generating good critiques over poor ones which still agree with the ground truth?

What happens if you train an LLM to optimise the resulting reward model, especially on prompts not in the initial instruction distribution of the LLM-as-a-Judge?
Does this policy LLM then outperform LLMs fine-tuned on different sources of reward?
This seems like a very important test to run to properly verify the method.

Do you know or have any idea what happens if you include response pair construction or instruction selection as part of the iterative process?

How does the proposed method avoid model collapse, as is implied to eventually happen with these iterative recursive schemes by Shumailov et al. (2024, https://www.nature.com/articles/s41586-024-07566-y)?

Multiple seeds and computing standard error/deviation would help discern the signal from the noise in regards to whether the iterative nature of the method provides significant, consistent benefits.
The reviewer notes that this might not be possible due to computational requirements involved.

In table 2, it's not clear whether the GPT4 baseline model was using 32 sample majority vote.
Please can you clarify this, as if it is not, the results of this table seems to be misleading and making an unfair comparison.

---

> ### Author Response · Authors · 2024-11-28
> **Thank you for your comments.**
>
> > Errata: Thanks for spotting these errata.
>
> We have fixed all of them. Please see the updated paper.
>
> > Overfitting on instructions and responses
>
> While we acknowledge the risk of overfitting with iterative training, our approach differs in that the instruction and response pairs remain constant, but the training data has new preference pairs in each iteration. These preference pairs include chain-of-thought reasoning and final conclusions, which introduces additional diversity and complexity to the training data. This can help alleviate the overfitting issue to some extent, as the model is exposed to a wider range of preferences and reasoning patterns, encouraging it to generalize better and adapt to new information.
>
> > Since the model is being fine-tuned to re-produce its own previous outputs (with some filtering applied), how can it learn to generate better critiques? Where is the exploration coming from, and why would it favor generating good critiques over poor ones which still agree with the ground truth?
>
> Our model iteratively improves its critique generation through a filtering mechanism, similar to rejection sampling methods used in large language models like Llama3.1. This approach ensures that the model is trained on high-quality data that aligns with the ground truth. Poor critiques are filtered out, which in our setting means those that incorrectly predict the better response. By refining its training data, the model can produce more accurate and reliable critiques over time.
>
> > Iterated training does not always help.
>
> Our experiments with RewardBench revealed a competition between categories, where improvements in one category (e.g., "chat") may come at the expense of another (e.g., "chat-hard"). This is likely due to different categories requiring distinct capabilities. Additionally, we observed fluctuations in performance across iterations, which can be attributed to the limitations of self-iterative methods and the model's capabilities given the seed training data.
>
> > From table 1, the method is beaten both overall and in the "Safety" and "Reasoning" categories by other methods.
>
> Our model outperforms all LLM-as-a-judge baselines in Table 1, achieving an overall performance of 88.3. In comparison, fine-tuning a LLM-as-a-judge model with the same data only reached 85.6. While we surpass Gemini 1.5 Pro overall, we lag behind in the reasoning category, possibly due to our starting point being llama3-70B instruct, which is not strong in reasoning.
>
> > The main motivation behind training a reward model is to provide a reward signal to optimise a downstream LLM's outputs to better reflect human values. The proposed method has not been evaluated on its ability to provide such a reward signal.
>
> Such analysis would be interesting to see. On the other hand, unlike reward models that only output scalar numbers or rankings, our model generates an evaluation plan (CoTs) that outlines how to make a judgment. These CoTs can be utilized in various ways, such as refining the original model response, analyzing model mistakes, understanding model behavior, and more.
>
> > What happens if you train an LLM to optimise the resulting reward model, especially on prompts not in the initial instruction distribution of the LLM-as-a-Judge?
>
> A comparison between LLM-as-a-judge and reward models would be insightful. Beyond performance, LLM-as-a-judge offers a unique advantage: it provides not only the decision but also the reasoning steps behind it, offering transparency and insight into the decision-making process. This can be valuable in various applications.
>
> > Do you know or have any idea what happens if you include response pair construction or instruction selection as part of the iterative process?
>
> This is not included as part of the iterative process.
>
> > How does the proposed method avoid model collapse, as is implied to eventually happen with these iterative recursive schemes by Shumailov et al. (2024, https://www.nature.com/articles/s41586-024-07566-y)?
>
> We haven't observed model collapse, possibly due to our approach of randomly sampling high-quality chain-of-thought judgments from previous iterations. This may introduce sufficient diversity and complexity into the training process, helping to prevent model collapse.
>
> > Multiple seeds and computing standard error/deviation.
>
> Thank you for your understanding. This is not feasible given our current computational resources.
>
> > In table 2, it's not clear whether the GPT4 baseline model was using 32 sample majority vote.
>
> It's worth noting that the GPT4 baseline in Table 4 did not utilize 32 sample majority votes. When we remove majority voting from our results, our performance drops to 78.9, which is very close to the GPT4-0125 score of 79.1. Therefore, we claim that our model achieves on-par performance with GPT4.

---

> > ### Comment · Reviewer_ubyr · 2024-11-30
> >
> > Thank you for the updated paper and responses to my concerns.
> >
> > Overfitting:
> > I'd already be / am mildly concerned about RLHF / finetuning methods which only train on a fixed set of prompts, even if new responses are generated to be trained on.
> > Given this method additionally fixes the responses it's just going to make things worse and even more brittle.
> > How can it get better over time as an evaluator if what its evaluations are being tested on remains fixed?
> > Surely it can only ever learn to overfit harder the more iterations you run?
> > I understand the CoT reasoning and conclusions will change, having some regularising effect, but I don't think this is enough.
> >
> > Exploration:
> > As written, only incorrect critiques are filtered out (lines 155/156).
> > Thus there is no optimisation pressure to produce as good critiques as possible, only ones that are at least correct in their final judgement.
> > Thus well-reasoned critiques which demonstrate good analysis are not favoured over poorly reasoned but (perhaps spurriously) correct ones.
> > Thus the model is not trained over time to produce better and better reasoning as long as it replicates the correct final judgement on the prompt/response pair, which is fixed and it has seen / been trained on many times before!
> >
> > To tie together the above points, what's stopping the model from memorising the correct answers to the prompt/response pairs whilst generating totally garbage reasoning that will not at all generalise to any other prompt/response?
> > I understand such memorisation and bad reasoning is not guaranteed, or even likely to happen, but it does seem poor that the method appears to do nothing to try and steer away from this outcome.
> > If I am incorrect and this is not the case, please clarify.
> >
> > "Our model outperforms all LLM-as-a-judge baselines in Table 1, achieving an overall performance of 88.3. In comparison, fine-tuning a LLM-as-a-judge model with the same data only reached 85.6. While we surpass Gemini 1.5 Pro overall, we lag behind in the reasoning category, possibly due to our starting point being llama3-70B instruct, which is not strong in reasoning."
> > These are good points, apologies for not noticing this when I originally read the paper.
> >
> > "It's worth noting that the GPT4 baseline in Table 4 did not utilize 32 sample majority votes. When we remove majority voting from our results, our performance drops to 78.9, which is very close to the GPT4-0125 score of 79.1. Therefore, we claim that our model achieves on-par performance with GPT4."
> > This ought to have been clearer in the paper.
> >
> > Additional experiments:
> > Whilst I understand the lack of computational resources, the strength of this paper is still limited by a lack of testing the trained reward models on their ability to provide a reward signal for an LLM to optimise.
> > The other benefits of CoT judgement you have outlined are interesting, but without a case study or experiment showcasing the benefits of this in practice, this is mostly speculation.
> >
> > Overall, whilst the authors have clarified some things, the majority of my concerns remain, and I maintain my score.

---

### Official Review · Reviewer_FjrG · 2024-10-30

**Soundness:** 2
**Presentation:** 2
**Contribution:** 4
**Rating:** 5
**Confidence:** 4

**Summary:**

The paper proposes a novel approach, "Self-Taught Evaluator," which enhances model-based evaluation without human annotations by using synthetic data only. The iterative self-improvement process generates contrasting outputs, curating training data from WildChat for a large language model (LLM) to improve as a judge. The authors demonstrate that this method can improve Llama-3-70B-Instruct from 75.4% to 88.3% accuracy on RewardBench, matching performance levels typically achieved with human-annotated data.

**Strengths:**

1. **Novel Contribution**: The paper addresses a significant challenge in model evaluation by eliminating the need for costly human annotations, which can be both expensive and quickly outdated.

2. **Technical Implementation**: The proposed solution is an end-to-end pipeline that includes data curation, iterative synthetic data generation, and model training. The training produceure is also properly outlined.

3. **Performance**: The method shows decent performance, competing with human-annotated evaluation models and LLM judges such as GPT-4-Turbo-125.

4. **Scalability and Cost-effectiveness**: This approach offers a scalable and more affordable alternative for model evaluation, which is particularly valuable for model developers.

**Weaknesses:**

## Methodological Concerns

**Reliability of LLM-as-a-Judge**: The paper does not thoroughly validate the reliability of various components in the LLM-as-a-Judge system used throughout the pipeline, raising concerns about the accuracy and consistency of judgments.

> Line 230: To perform prompt selection, we annotate the category of each instruction with the Mixtral 22Bx8 Instruct model, using the template in Figure 7 and select 20,582 examples in the reasoning category, as we expect these to be challenging inputs.

1. *Selection and Justification of Categories*: It is unclear whether the reasoning category chosen is specific to “Knowledge and Reasoning” or includes others like "Coding" or "Social Studies." Clarifying which categories were deemed “challenging” and why could strengthen the rationale behind the selection process.

2. *Assumption of Prompt Difficulty*: The choice to prioritize the “reasoning” category as the most challenging is not fully justified. Other categories like “Coding” or “Social Studies” could also be complex; a justification for the choice of the reasoning category would enhance the paper’s clarity.

3. *Lacks Validation of Categorical Classification*: The study's reliance on Mixtral 22Bx8 as a judge for classification raises concerns about potential biases and reliability. Without proper validation and ablation studies, the accuracy of the classification process and the quality of selected prompts remain questionable. Including accuracy metrics or a confusion matrix would significantly strengthen the claims made about classification quality. Ablation on using different LLMs as judge is also desirable.

> Line 126: We use the following prompt template which is used to generate a 'worse response' y^l. Given an instruction x and baseline response y^w generated by an instruction-following LLM as usual, this prompt is used to first generate a 'noisy' version x′ of the original instruction x, and then a best-attempt y^l at responding to x′. y^l is then treated as a poor response to x, giving a preference pair y^w_i ≻ y^l_i.

4. *Assumption of Poor Response Quality*: While the methodology treats responses from certain prompts as “worse,” the inherent biases of LLMs and potential quality variances raise questions about consistency.
5. Correct me if I am mistaken, I cannot find examples for the generation of “worse responses” anywhere in the paper, including the appendix. Examples would be helpful for further illustration.

> Line 244: At inference time when evaluating final performance, we sample generations N times and take the final judgment to be the most common verdict.

6. *Majority Vote Methodology*: The approach of using a majority vote to finalize judgments could benefit from additional clarification, specifically on how judgments are consolidated and what impact this has on final model performance.

> Line 250: To understand the effectiveness of the proposed method, we generate synthetic judgments using the same approach but based on the following data sources.

7. *Synthetic Data Generation Steps*: The paper proceed to explain high-level steps for synthetic curation, however, lacks sufficient detail on the exact generation steps, limiting reproducibility.

8. *Data curation details*: It is unclear how exact the data is being curated. Starting with WildChat dataset, how many conversations were applied using the prompt selection procedure? Did the author deduplicate the dataset? Did author ran PII detection? Were multi-turn conversation being included? If multi-turn conversation are being include, how are they handled during prompt selection and judgment annotation (eg. concat all the turns)? Does the curated dataset include non-English conversation? Addressing these question and provide a detailed explanation will greatly improve reproducibility of the work.

## Analysis Concerns

> Line 497: We further instruct Llama-3-70B-Instruct to infer the complexity (using a score of 1–5) and category of each input instruction.

9. *Validation of Complexity Inference*: Relying on Llama-3-70B-Instruct to infer prompt complexity raises questions regarding its ability to accurately gauge task difficulty. Validation experiments could bolster the validity of the complexity categorization.

10. *Presence of Simple Prompts in Curated Data*: Despite aiming to filter for challenging prompts, many simpler prompts remain, suggesting possible gaps in the selection methodology. Additional validation could enhance prompt filtering accuracy.

## Suggestions for Improvement

1. **Validation of Pipeline Components**
   Additional validation experiments are needed for components like prompt selection and judgment annotation, particularly since both stages heavily rely on LLM-based evaluators.

2. **Ablation Studies**
   The paper would benefit from ablation studies to justify choices, such as using Mixtral-22Bx8 for prompt selection, as well as alternative configurations that might affect the pipeline’s effectiveness.

3. **Generalizability**
   Demonstrating the approach’s adaptability to other models would help establish broader applicability and enhance its value to the field.

4. **Bias Analysis**
   A thorough analysis of potential biases, including length bias or stylistic preference, in the trained evaluator would clarify whether the approach mitigates or exacerbates common biases in LLM evaluation.

5. **More baselines**
  The paper compare trained evaluator against gpt-4-turbo-0125 on MT-Bench. The result can be strengthen by using more than one baseline and include other SOTA LLM judges. Otherwise, it is also unclear whether gpt-4-turbo-0125 is actually a good LLM judge.

**Questions:**

1. **Validation of Prompt Selection and Evaluation**
   Could you provide further clarity on the categories chosen for “reasoning” and explain why they were deemed the most challenging, specifically over alternatives like “Coding” or “Social Studies”?

2. **Judgment Selection Consistency**
   In the data collection process, was there a standardized method to ensure that “worse responses” generated were indeed inferior? Examples of these responses, especially in the appendix, would improve understanding of this process.

3. **Majority Vote Implementation**
   How was the majority vote for final judgments implemented? What impact did this have on the model’s improvement scores?

4. **Synthetic Data Generation Process**
   Could the authors provide more concrete steps for generating synthetic judgments based on additional data sources? This would aid in replicability.

5. **MT-Bench Experiment**
  Since MT-Bench is multi-turn, did the author observe any difference in the performance of the evaluator on first turn versus the second turn?

---

> ### Author Response · Authors · 2024-11-28
> **Thank you for your comments.**
>
> First we will go over questions, and then over suggestions for improvements:
>
> > Validation of Prompt Selection and Evaluation
>
> It happens that the reasoning category is rather a vague term within LLM benchmarks. For example, coding and math prompts are part of the reasoning category in the rewardbench such as math-prm, hep-cpp, hep-go, hep-java, hep-js, hep-python, hep-rust datasets. This is why we consider this to be more challenging compared to other prompts such as creative writing and other chat-like prompts.
>
> > Judgment Selection Consistency
>
> We included examples of inferior responses (Table 15 in the Appendix). In addition, all the data is planned to be released to analysis and use by the community. We did not make an extensive verification of the worse response quality w.r.t., a better one since we wanted to reduce the amount of human annotation signal that we might inject in this research design.
>
> > Majority Vote Implementation
>
> Majority voting is implemented based on the fact that the final verdict (A vs B) is parsable. That is, judgements are generated N times, N verdict signals are extracted, and the final verdict corresponds to the most common one following the counts. This improves our model from 88.3 to 88.7 on RewardBench, as shown in Table1.
>
> > Synthetic Data Generation Process
>
> We included some details about generations in Section 4.2. We will share our data as well as our code for data generation on Github.
>
> > MT-Bench Experiment
>
> We had only used 1st turn data for MTBench evaluation since our training data does consider any multi-turn conversations. We will make this clear in our evaluation description!
>
> Now we go over improvement suggestions:
>
> > Validation of Pipeline Components and Ablation Studies
>
> We agree that more validation signals would help, but the best validation signal is the performance we got using the resulting model. Due to time constraints, we won’t be able to add more validation experiments, and we think it does not limit the overall results delivered in this work. As per ablation studies, we have included multiple ablations about different data sources that is one of the foundational pipeline settings (the source of prompts). We believe future work might be coming from ablating this pipeline and proposing more advanced versions of it.
>
> > Generalizability
>
> We hope that our pipeline will be attractive enough and tested with other model families by other researchers.
>
> > Bias analysis
>
> This is a valid point. We believe this direction is crucial and warrants our attention, even though it's not currently reflected in popular benchmarks. In this paper, although we haven't conducted specific bias analysis, we did try to measure position-consistent accuracy: the accuracy when the model consistently makes correct predictions, even when swapping the order of two response options. This metric is reported on the HelpSteer validation split in Table 3.
>
> > More baselines
>
> We used mt-bench evaluators following their guidelines to keep our numbers reproducible w.r.t., the other results reported in literature. For rewardbench we have other baselines such as gemini model from Google and larger size llama models. We also compare to other LLM-as-a-judge models (most of them came out after our submission) as shown in Table 14 in the Appendix.
>
> We hope our answers and manuscript revisions (please see updated PDFs with blue colored revisions) convince you to adjust your score. Thank you!

---

> > ### Comment · Reviewer_FjrG · 2024-12-03
> > **Official Response from the Reviewer**
> >
> > Thank the authors for their response to my questions and comment.
> >
> > I understand the difficulty to completely fix some of the limitation I bought up in my reviewer. The lack of validations to certain components of the paper remains a weakness, but I understand the authors' best ability to address them.
> >
> > I will maintain my score.

---

### Official Review · Reviewer_7PWJ · 2024-11-01

**Soundness:** 3
**Presentation:** 4
**Contribution:** 3
**Rating:** 5
**Confidence:** 4

**Summary:**

This paper proposes a new way to train the evaluators for the capabilities of LLMs which relies on only synthetically generated data. The authors start with a set of user instructions which are classified and sampled to represent different categories. Then, they generate pairs of responses where one is constructed to be better than the other. First, they generate a response from the model which is going to represent a better response. Second, they prompt a LLM to modify the initial user task to be semantically similar but not identical and they generate the response for that task and this sample is going to represent a worse response in the pair. For each pair, they sample N reasoning traces and judgements with LLM-as-a-Judge and select a trace which agrees with the synthetically constructed order of the samples. This trace is then used for finetuning the model. They repeat this process iteratively by using better and better LLM-as-a-Judge models but finetuning the same original model. The experiments show that on several benchmarks their method results in a better model than the same initial model trained on human data.

**Strengths:**

- This paper presents an efficient method for an important problem of LLM evaluation. The method relies on synthetic data without the use of human data and thus it is easier to scale in practice.
- The presentation of the paper is very clear and it is easy to follow and find relevant information
- The proposed method is quite original, especially the component on generation of artificial pairs of responses where one element is constructed to be better than the other element.
- The experimental section contains comprehensive experiments showcasing the benefits of synthetic data over human data. The results are quite encouraging and have a potential to have an impact in the community.

**Weaknesses:**

- My main concern is about using *two* language models in the experiments. While the method is presented as a "self-taught" evaluator, the actual experiments rely on the use of two models: Mistral 22Bx8 Istruct for generating the initial synthetic responses, initial judgments and categorizing the queries, and Llama3-70B-Instruct for everything else. How is the need for the second model motivated? In this case, wouldn't it be the situation of distilling the knowledge of two LLMs into one rather than being self-taught?
- Regarding the experimental section and the baselines, the authors mentioned some related work that sounds to be highly applicable in the studied setting, such as the Best-of-N method. Were there any attempts to compare against it?
- The proposed method contains many steps / components that contribute to the performance of the policy. While there are several ablations and comparisons presented in the experimental section, it would be nice to understand in which degree each component affects the solution.
- In table 2 it seems that the performance on iteration 5 is worse than on iteration 4. Why is this the case? What is happening here?

Some minor questions and clarifications:
- When comparing against the human data, were the sizes of the datasets identical? Was the user instructions with human labels filtered in the same way (which seems to preserve more complex instructions) as the user instructions used to generate the synthetic pairs?
- In terms of formatting, I think putting brackets around citations would make it easier to read the text.
- Line 090: West->Best?
- For the understanding of the algorithm, I would like to see some examples of the "related but different" prompts and to see in what kind of answers they result. Intuitively, it is hard to make difficult comparison examples synthetically, and I would be interested to read examples from this approach
- What proportion of the traces agree with the synthetic labels over the training iterations?
- Details about the model selection (line 225-227 are not very clear)
- As far as I understand, for the final inference, there are N samples generated and the majority vote is performed for the proposed method (line 244-247). Is the same done for the baselines?
- What is the conclusion from Table 6? What combination to use, if any?

**Questions:**

- I would like to understand the motivation behind using two language models for different steps in the experiments. Why is it necessary? What would happen if the same model is used everywhere? How does the fact of using two models reflect on positioning the method as self-taught? How the use of an additional model should be reflected in the baselines in order to make a fair comparison?
- I would like to understand the importance of various components of the proposed method and how it compass to other existing synthetic data methods like the Best-of-N mentioned in the related work section.

---

> ### Author Response · Authors · 2024-11-28
> **Thank you for your comments.**
>
> > The use of Mistral 22Bx8 Instruct for generating the initial synthetic responses.
>
> We utilized the Mixtral model to generate initial synthetic responses due to our existing infrastructure, which enabled rapid inference with this model. However, we emphasize that this choice is not necessary. As demonstrated in the Appendix (Table 10), replacing the Mixtral model with llama3.0 models for generating synthetic responses or judgements yields similar results, highlighting the flexibility of our approach.
>
> > Regarding the experimental section and the baselines, the authors mentioned some related work that sounds to be highly applicable in the studied setting, such as the Best-of-N method. Were there any attempts to compare against it?
>
> For a detailed comparison with competing models, please refer to Table 14 in the Appendix. It is important to note that these competing models are built using a large amount of human-annotated datasets, whereas our model only requires synthetic datasets without any human labels.
>
> Regarding the "West-of-N" paper, they do not report any results on RewardBench and have not released their code. As a result, it is currently challenging for us to compare our model with theirs. However, we will make every effort to provide a comparison whenever they share their numbers or code.
>
> > The proposed method contains many steps / components that contribute to the performance of the policy. It would be nice to understand in which degree each component affects the solution. In table 2 it seems that the performance on iteration 5 is worse than on iteration 4.
>
> We identify high-quality synthetic data and iterative training as the two most critical components of our model. The effectiveness of synthetic data is demonstrated in Table 4, where we conduct ablation experiments with different datasets. Similarly, the benefits of iterative training are evident in Table 1, which shows that performance improves from 83.9 (first iteration) to 88.3 (5th iteration). Notably, we observe that further iterations beyond this point result in only marginal improvements, suggesting that the model may have reached an upper bound with the given training data under our iterative training framework.
>
> > When comparing against the human data, were the sizes of the datasets identical? Was the user instructions with human labels filtered in the same way (which seems to preserve more complex instructions) as the user instructions used to generate the synthetic pairs?
>
> The size of our datasets can vary slightly from one iteration to another. This is because we use the model from the previous iteration to generate new reasoning traces and judgments, and then select the correct ones. We did not apply the same filtering process to datasets with human labels, as these datasets are already carefully processed and filtered by their authors, and are therefore expected to be of relatively high quality.
>
> Another reason for this approach is to provide a fair comparison between our model and the baselines. By using datasets with human labels in the same way as other papers, we ensure that our results are directly comparable to those reported in the literature.
>
> > Putting brackets around citations would make it easier to read the text.
>
> Thanks for pointing this out! We have fixed it.
>
> > Line 090: West->Best?
>
> It’s actually “West”, please check out this: https://arxiv.org/html/2401.12086v2
>
> > For the understanding of the algorithm, I would like to see some examples of the "related but different" prompts and to see in what kind of answers they result.
>
> Please see the examples included in the appendix (Table 15).
>
> > What proportion of the traces agree with the synthetic labels over the training iterations?
>
> This is a good question, and we haven't conducted such an analysis yet. Intuitively, we would expect the agreement to increase over training iterations as the model becomes more refined and accurate.
>
> > Details about the model selection (line 225-227 are not very clear)
>
> For model selection, we utilize a metric that aggregates two types of accuracies:
> 1. Accuracy when the model makes a correct prediction.
> 2. Accuracy when the model consistently makes a correct prediction even when swapping the order of two response options.
>
> > There are N samples generated and the majority vote is performed for the proposed method (line 244-247). Is the same done for the baselines?
>
> No, this is not the case for baselines. As evident from Table 1, majority voting does not provide significant improvements (88.3->88.7), indicating that it may not be an effective approach in this context.
>
> > What is the conclusion from Table 6?
>
> Mixing data for building a Large Language Model (LLM) as a judge is not a straightforward process. Although this result may be considered negative, we believe it's essential to share our findings with the research community to contribute to the ongoing discussion and exploration of LLMs.

---

### Official Review · Reviewer_iCDS · 2024-11-02

**Soundness:** 3
**Presentation:** 2
**Contribution:** 3
**Rating:** 5
**Confidence:** 3

**Summary:**

This paper presents a novel approach to training LLMs to act as **evaluators**, or "judges," without relying on human-labeled data. The key method involves a self-supervised, iterative training process where the model generates its own **synthetic data** and **preference labels**, using these to refine its evaluative skills over multiple rounds, resulting in curriculum learning. This process allows the model to progressively improve its judgment abilities without external human supervision. In tests on benchmarks like RewardBench and MT-Bench, the self-taught evaluator achieves competitive, sometimes superior, results compared to reward models trained with human annotations.

**Strengths:**

- The self-taught framework presented in this paper is a quite novel. It uses an iterative self-improvement process that enables the model to independently refine its judgment skills, offering valuable insights for future research in self-supervised evaluation methods for LLMs.

- The paper is well-structured and clearly written.

- The proposed self-taught method could effectively reduces the dependency on human labeling by enabling the model to generate its own synthetic data and preference labels. It offers a scalable solution for training evaluation models for LLMs.

- The empirical results are highly promising. On competitive benchmarks like RewardBench and MT-Bench, the self-taught models match or even exceed the performance of traditional reward models that rely on human annotations, highlighting the method’s strong potential for practical applications.

**Weaknesses:**

- The method has only been tested on one specific LLM variant (LLAMA3-70B-Instruct), making it unclear whether the approach generalizes well to other types of LLMs or models of different sizes and architectures.

- The process for generating contrasting synthetic preference pairs is relatively simple and lacks refinement. The prompt used to generate suboptimal responses often results in fairly static patterns, and it remains unverified whether the generated response $y^l$ is indeed worse than the original. There is limited discussion on corner cases or situations where synthetic data quality might be compromised. A more carefully crafted design for preference pairs could improve the model’s ability to distinguish subtle judgment differences.

- The paper provides limited comparison with other LLM-as-a-judge methods, relying primarily on comparisons with GPT-4-0125 while omitting other competitor models from existing literature in LLM-as-a-judge frameworks.

- The computational complexity of the iterative self-taught evaluation process is not fully discussed. Generally, this approach involves a curriculum learning process requiring multiple iterations, and further discussion on the computational demands and any associated performance gains would strengthen the analysis.

- In practice, determining the correctness of a judgment can be challenging. The proposed method relies on a fairly straightforward approach for judgment annotation, and it’s uncertain whether this consistently leads to high-quality judgments. The quality of judgments has not been independently verified.

- While the performance of the self-taught evaluator generally improves with each iteration, there are scenarios (e.g., reasoning ability for RewardBench doesn't increase as more iterations are gone through) where performance declines. More in-depth analysis and discussion on these cases would provide valuable insights, and investigating a potential upper bound for the model’s performance could also be beneficial.

- The paper lacks clarity on the amount of synthetic data needed per iteration and how performance might vary with different volumes of synthetic data. Providing guidelines on optimal data requirements for each iteration would improve the method’s practical applicability.

**Questions:**

-Please use \citep in place of \cite where appropriate to ensure citation consistency.

- I would consider revising the score if the paper includes a more comprehensive discussion of potential drawbacks, along with a comparison to other LLM-as-a-judge baselines.

**Details Of Ethics Concerns:**

N.A

---

> ### Author Response · Authors · 2024-11-28
> **Thank you for your comments.**
>
> >I would consider revising the score if the paper includes a more comprehensive discussion of potential drawbacks, along with a comparison to other LLM-as-a-judge baselines.
>
> Thank you for your kind comments. We have taken your feedback into consideration and enhanced the Limitations section to provide a more in-depth discussion of potential drawbacks. Additionally, we have included comparisons to other LLM-as-a-judge baselines and experiments on different models above, which are also temporarily appended at the end of the Appendix (Table 13, 14) for reference. Please refer to the updated paper for these changes.
>
> In Table 13, we applied our approach to the LLaMA2, LLaMA3, and LLaMA3.1 models. Due to time constraints, we only present results after the first iteration of supervised fine-tuning. Our approach consistently demonstrates performance improvement across different models, even with just one iteration.
>
> In Table 14, we present a comparison between our Self-taught evaluator and several other LLM-as-a-judge models. The numbers are taken from the RewardBench Leaderboard. Note some of the models came out after the submission of our paper. The state-of-the-art (SOTA) performance is achieved by [1], where they fine-tune the Llama-3.1-70B instruct model on a pool of various human-labeled preference datasets, totaling 80K pairs. The remaining models (except for our Self-taught evaluator) are built on top of different base models but all rely on human-labeled preference datasets. In contrast, our Self-taught evaluator is based on the Llama-3.0-70B instruct model and only 10k synthetic pairs. Despite this, it still achieves good performance, demonstrating its effectiveness as an evaluator.
>
> [1] (Tu et al., 2024) Skywork Critic Model Series.
>
> > A more carefully crafted design for preference pairs could improve the model’s ability to distinguish subtle judgment differences.
>
> Thanks for suggesting! The focus of this paper is trying to present a framework for building powerful LLM-as-a-judge models by using high quality synthetic datasets. We plan to further improve our model by refining our synthetic data creation pipeline as you suggested here.
>
> >  Further discussion on the computational demands and any associated performance gains.
>
> Our approach inherently requires additional computational resources due to the multiple iterations of training and evaluation on validation sets. To address this, we plan to optimize the efficiency of iterative training in our future work and provide a detailed analysis of the computational demands involved.
>
> > The quality of judgments has not been independently verified.
>
> Determining the correctness of judgments remains a significant challenge. Investigating methods to derive accurate judgments is an intriguing research question that warrants further exploration. We believe that high-quality judgments have the potential to be even more beneficial, as our straightforward approach has already demonstrated promising results.
>
> > While the performance of the self-taught evaluator generally improves with each iteration, there are scenarios (e.g., reasoning ability for RewardBench doesn't increase as more iterations are gone through) where performance declines.
>
> Through our experiments with RewardBench, we observed a phenomenon where different categories competed with each other. Specifically, we found that the performance of one category could improve at the expense of another. For example, the "chat" category might show improved performance while the "chat-hard" category experiences a decline. We speculate that this competition arises from the fact that different categories require distinct capabilities. The "chat" category focuses on stylistic and writing tone aspects, whereas the "chat-hard" category prioritizes factuality and correctness. This competition may contribute to performance declines in certain scenarios.
>
> > The paper lacks clarity on the amount of synthetic data needed per iteration and how performance might vary with different volumes of synthetic data.
>
> We used about 10K synthetic data examples in our experiments. We have included such details.  Please see the updated paper.
> > Please use \citep in place of \cite where appropriate to ensure citation consistency.
>
> We have fixed this. Please see the updated paper.

---

### Author Response · Authors · 2024-11-28

We appreciate the time and effort taken by all reviewers to provide constructive comments. We have carefully addressed your feedback and updated our paper accordingly. Please refer to our detailed responses to your specific questions, as well as the revised paper.

---

### Meta-Review · Area_Chair_yuBK · 2024-12-21

**Metareview:**

**summary**

The paper introduces Self-Taught Evaluator for training LLMs as evaluators without relying on human annotations, using a fully synthetic, iterative self-training process. The method generates contrasting response pairs for given instructions, curating synthetic preference labels to train the model in an iterative curriculum learning approach. By leveraging CoT reasoning and progressively improving the model’s evaluative capabilities through self-supervised fine-tuning, the approach demonstrates substantial performance improvements on benchmarks like RewardBench and MT-Bench.

**strengths**

* By enabling models to generate their own synthetic data and preference labels, the method significantly reduces the need for human labeling, offering a scalable solution for training evaluators.
* The method demonstrates strong performance on benchmarks like RewardBench and MT-Bench, matching or surpassing traditional models reliant on human annotations, which underscores its potential for practical application.
* The paper is well-structured and clearly written, making it accessible and easy to follow.

**weaknesses**

* Judgment quality assurance: As pointed by several reviewers, the method's reliance on generated judgments without independent validation raises concerns about the accuracy and reliability of these judgments, which are crucial for training the evaluative models.

**decision**

This paper introduces an innovative method that employs an iterative self-improvement process to enhance the judgment skills of LLMs. Considering the impacts/importance of LLM evaluators, I think this paper provides valuable insights to the community. However, there are several areas that require further exploration. Notably, addressing the concerns regarding judgment selection consistency is crucial. While the use of synthetic preferences is a significant advantage of the proposed method, the lack of independent validation for judgment evaluation raises questions about the trustworthiness and actual quality of the models. Although the authors have demonstrated the quality of their method on standard benchmarks, this does not necessarily guarantee its correctness. This issue is particularly critical given that one of the primary uses of LLM evaluators is to ensure the alignment of LLMs with human values.

**Additional Comments On Reviewer Discussion:**

Reviewers raised several concerns and questions regarding various aspects of the paper, such as computational complexity, design choices, and baseline comparisons, to which the authors provided reasonable responses for most. However, the responses regarding the assurance of judgment quality seemed insufficient. The authors aim to minimize the amount of human annotation to reduce potential biases in the research design, but this step is crucial for validating the effectiveness of their work. Ensuring high-quality judgment in models, especially when reducing human oversight, is essential to substantiate the claims made in this study.

---

### Decision · Program_Chairs · 2025-01-22

Reject